# Technological Development and Application of Plant Genetic Transformation

**DOI:** 10.3390/ijms241310646

**Published:** 2023-06-26

**Authors:** Wenbin Su, Mingyue Xu, Yasmina Radani, Liming Yang

**Affiliations:** State Key Laboratory of Tree Genetics and Breeding, Co-Innovation Center for Sustainable Forestry in Southern China, Nanjing Forestry University, Nanjing 210037, China; issuwb@outlook.com (W.S.); mingyuexu0116@163.com (M.X.); radani.yasmina@gmail.com (Y.R.)

**Keywords:** plant genetic transformation, *Agrobacterium*, particle bombardment, nanoparticles

## Abstract

Genetic transformation is an important strategy for enhancing plant biomass or resistance in response to adverse environments and population growth by imparting desirable genetic characteristics. Research on plant genetic transformation technology can promote the functional analysis of plant genes, the utilization of excellent traits, and precise breeding. Various technologies of genetic transformation have been continuously discovered and developed for convenient manipulation and high efficiency, mainly involving the delivery of exogenous genes and regeneration of transformed plants. Here, currently developed genetic transformation technologies were expounded and compared. *Agrobacterium*-mediated gene delivery methods are commonly used as direct genetic transformation, as well as external force-mediated ways such as particle bombardment, electroporation, silicon carbide whiskers, and pollen tubes as indirect ones. The regeneration of transformed plants usually involves the de novo organogenesis or somatic embryogenesis pathway of the explants. Ectopic expression of morphogenetic transcription factors (*Bbm*, *Wus2,* and *GRF-GIF*) can significantly improve plant regeneration efficiency and enable the transformation of some hard-to-transform plant genotypes. Meanwhile, some limitations in these gene transfer methods were compared including genotype dependence, low transformation efficiency, and plant tissue damage, and recently developed flexible approaches for plant genotype transformation are discussed regarding how gene delivery and regeneration strategies can be optimized to overcome species and genotype dependence. This review summarizes the principles of various techniques for plant genetic transformation and discusses their application scope and limiting factors, which can provide a reference for plant transgenic breeding.

## 1. Introduction

Plant genetic transformation is an important pathway to improve plant yield, quality, and tolerance to abiotic/biotic stress [1]. There are numerous proven genetic transformation methods that can stably introduce new genes into the nuclear genomes of different plant species. However, despite decades of technological advancement, efficient plant transformation and regeneration remain a challenge for many species [2]. Plant genetic transformation is mainly divided into two steps: biomolecule delivery and transgenic plant regeneration. The main bottleneck in successful plant genetic transformation is how biomolecules enter plant cells through the hard multi-layer cell wall and the subsequent regeneration of transgenic plants from an in vitro cultured explant, either via de novo organogenesis or somatic embryogenesis [3].

Exogenous genes can be delivered to plant cells by *Agrobacterium*, particle bombardment/gene gun, electroporation, the pollen tube pathway, and other mediated delivery methods [4,5]. However, these methods have multiple drawbacks. For example, *Agrobacterium*-mediated delivery is limited by species genotype and explant dependence. Particle-bombardment-mediated delivery frequently causes chaotic DNA integration events [6] and plant tissue damage, rendering regeneration inefficient [7]. So far, there is no plant genetic transformation method that can deliver various biomolecules to a wide range of plant genotypes and species without the use of external force and tissue damage [8]. In recent years, researchers have become more interested in the delivery of biomolecules via nanomaterials. These nanoparticles can enter plant cells on their own, and dicots and monocots show different degrees of direct absorption of various types of nanoparticles, including magnetic nanoparticles, peptide nanoparticles, layered double hydroxide nanosheets, DNA nanostructures, and carbon nanotubes [1,9]. Compared to traditional gene delivery methods, nanoparticle-mediated delivery has the advantages of directly crossing biomembranes, protecting and releasing multiple cargoes, and achieving multidimensional targeting through chemical and physical tunability [8].

Due to the difficulty in regenerating transformed plants from explants, researchers have developed many strategies to overcome the problems of genotype dependence and low transformation efficiency caused by limited regeneration ability during tissue culture, such as adding different hormones to the medium, using explants with less genotype dependence and ectopic expression of morphogenetic transcription factors (MTFs) [10]. The most widely used strategy is to enhance plant regeneration through ectopic expression of MTFs such as *Baby boom* (*Bbm*), *Wuschel2* (*Wus2*) [3,11,12], and *Growth*-*regulating factors* (*GRFs*) [13,14,15,16]. Lowe et al. showed that overexpression of *Bbm* and *Wus2* successfully transformed monocot genotypes or explants that were previously difficult to genetically transform [11]. Furthermore, *GRF4*-*GIF1* chimera from citrus and grape enhanced citrus plant regenerative capacity, suggesting that *GRF4*-*GIF1* chimera overexpression extends the range of convertible genotypes [14]. Overexpression or inducible expression of these MTFs not only increased the frequency of transformation but also expanded the range of convertible species and genotypes [10].

The development of an efficient genotype-independent plant transformation system is critical for translating advances in plant molecular biology into crop improvement [2]. In this review, we introduce commonly used plant genetic transformation techniques as well as recently developed flexible approaches for plant genotype transformation, and we discuss how to optimize gene delivery and regeneration strategies to overcome species and genotype dependence.

## 2. Techniques of Plant Genetic Transformation

Plant genetic transformation techniques are classified into two types: indirect genetic transformation and direct genetic transformation [17]. Indirect genetic transformation is one method that uses organisms as a vector, such as *Agrobacterium*-mediated gene transfer into target cells, whereas direct genetic transformation uses external forces to deliver target genes into plant cells, including particle bombardment/gene gun, electroporation, liposomes, silicon carbide, microinjection, and pollen-tube-pathway-mediated plant genetic transformation methods [4,5].

### 2.1. Indirect Genetic Transformation

The indirect genetic transformation method in plants mainly refers to *Agrobacterium*-mediated transformation. *Agrobacterium* species, including *Agrobacterium tumefaciens* and *Agrobacterium rhizogenes*, contain plasmids that induce tumors (Ti) or hairy roots (Ri) [18]. By modifying the plasmid, a segment of the T-DNA on the Ti/Ri plasmid can be transferred and integrated into the plant genome, and the target gene can be co-integrated with the T-DNA into the plant genome. The process of T-DNA formation transfer to plants cells is shown in Figure 1. Phenolics or acidic sugars released from the injured part of the plant are sensed by VirA, which then activates VirG via phosphorylation. VirG further induces the expression of the *Vir* (*Virulence*) gene in *Agrobacterium tumefaciens*. Following that, the combined action of induced VirD1 and VirD2 cleaves the Ti/Ri plasmid’s T-DNA region at the left border (LB) and right border (RB) repeat sequences. During cleavage, VirD2 is covalently attached to the 5’ end of the T-DNA. VirD2/T-DNA then leaves the bacteria via T4SS (Type IV secretion system). Furthermore, the single-stranded DNA-binding protein VirE2 may noncovalently coat the T-chain in plant cells, forming the VirD2/VirE2/T-DNA T-complex and promoting T-DNA integration into the plant genome [18]. Unlike *Agrobacterium tumefaciens* which harbors the Ti plasmid that induces tumors on host plants’ crown region, the hairy roots produced by the Ri plasmid carried by *Agrobacterium rhizogenes* exhibited multi-directional growth, multi-lateral roots, non-geotropism and rapid growth on the medium without any plant growth regulators [19,20]. Recently, Cao et al. used *Agrobacterium rhizogenes* to inoculate explants, generating transformed roots that produce transformed buds due to root suckering, thereby successfully achieving heritable transformation of plant species, including herbaceous plants (*Taraxacum kok-saghyz* and *Coronilla varia*), a tuberous root plant (sweet potato), and woody plants (*Ailanthus altissima*, *Aralia elata,* and *Clerodendrum chinense*) [21]. This method enables the transformation of some species [21], that were previously difficult to genetically transform, under non-sterile conditions and without the need for tissue culture.

*Agrobacterium*-mediated plant genetic transformation has the advantages of high efficiency, simple operation, and genetic stability, and it can be used to transform the vast majority of dicots and a few monocot plants (Table 1). However, many factors limit the successful application of *Agrobacterium*-mediated transformation in monocot plants. Compared to dicots, most monocots cannot be naturally infected by *Agrobacterium* because they have no obvious divided cell, limiting the genetic transformation of monocot plants mediated by *Agrobacterium* [1]. The breakthrough in monocot plant transformation by *Agrobacterium* comes from a better understanding of the key factors or parameters required for effective grain infection and gene transfer, such as using explants with a large portion of active dividing cells, which refers to immature embryos. Also, it has been found that employing highly toxic *Agrobacterium* strains and suitable vectors are necessary for *Agrobacterium* to successfully transform cereal [22]. In addition, efficient selection of stably transformed cells from a large number of non-transformed cells is an important part of successful transgenic plants’ transformation and regeneration, while the early dicots’ transformation system relied on aminoglycosides resistance, including kanamycin, neomycin and G418, which proved ineffective in most cereal crops [23]. Herbicide-resistant markers were used to select maize, wheat, and barley transformants [23]. The optimization of the vector [24], application of a hypervirulent *Agrobacterium* strain [25,26], as well as the use of suitable selectable markers [27] improved the efficiency of transformation mediated by *Agrobacterium* of monocot plants (Table 1).

### 2.2. Direct Genetic Transformation

#### 2.2.1. Particle-Bombardment-Mediated Transformation

Particle bombardment (also known as gene gun) is a physical method of introducing exogenous DNA directly into the plant genome [43,44]. Particle-bombardment-mediated plant transformation is not limited to the source of receptor materials; cells, calli, immature embryos, and organs can all be used as targets for transformation (Table 2). In this method, the target gene is coated on the surface of gold or tungsten powder to construct a DNA-coated microcarrier. High-pressure helium pulses accelerate the DNA-coated microcarrier into the gas acceleration tube using an electric discharge or a pressurized helium gas stream (Figure 2a). These particles gain sufficient momentum to pierce recipient cells at high speed, while the target gene coated on the outside remains in the cell [6,45] and is eventually integrated into the plant’s chromosome, producing the transformed plant [5] (Figure 2a). After *Agrobacterium*-mediated plant genetic transformation, particle bombardment has gradually become one of the most prominent transformation methods and has been successfully applied to many plant species (Table 2). Plant genetic transformation mediated by particle bombardment is distinguished by the diversity of target materials and ease of operation. However, when compared to *Agrobacterium*-mediated transformation, it has some disadvantages, such as a lower transformation rate, higher costs, and unprotected exogenous DNA [17]. In addition, this method can only transfer DNA fragments smaller than 10 kb because larger fragments are easy to break during bombardment or have weak adherence to metal particles, resulting in chaotic DNA integration events [6]. The DNA repair mechanism mainly includes a non-homologous terminal junction (NHEJ) for the cell nucleus only and homologous recombination (HR) or plastids and the cell nucleus; particle collision in the transformation process mediated by particle bombardment enables homologous sequences to lead to transcription or post-transcriptional gene silencing through DNA–DNA, DNA–RNA, and RNA–RNA interactions [1]. Researchers are investigating potential mechanisms for these complexities and seeking solutions, and targeted DNA insertion at suitable genomic sites in plants is a promising alternative [46]. Particle bombardment has been employed to co-deliver the CRISPR/Cas or ZFNs machinery and the repair template into plant tissues; targeted insertions of selectable marker genes by particle bombardment have been achieved in rice [47] and soybean [48] (Table 2).

#### 2.2.2. Electroporation-Mediated Plant Transformation

Electroporation is an electrical transformation method that uses short, high-field pulses to create transient pores in the plasma membrane of target cells, increasing the permeability of the host cell membrane [49,50]. Protoplast or cell and DNA are incubated together, and then short- and high-field pulses are used to generate transient pores in the membrane of the target cell (Figure 2b). The water pores formed by electric induction can be divided into two stages on the lipid bilayer. Water molecules first penetrate the bilayer, forming an aqueous pore. Second, the polar head groups of adjacent lipids are reoriented towards the water molecules, forming hydrophilic pores and allowing transmembrane transport of other impermeable molecules, thus introducing DNA into the recipient cell [51]. Under an optimal electrical pulse, these pores can be resealed, restoring the cells to their original state [50] (Figure 2b). Compared to *Agrobacterium* and particle-bombardment-mediated plant transformation, electroporation-mediated transformation has the advantages of rapid application, low cost, and a highly stable transformation rate [52]. In addition, unlike particle bombardment, which tends to introduce large plasmid concatemers, electroporation produces primarily single-copy plasmid fragments [53]. The main disadvantage of electroporation is the difficulty in transforming plant cells with thick cell walls [49], and it only works with a limited number of receptor species. Furthermore, strong electric field pulses can destroy the naked gene, resulting in inaccurate translation of the final product [1].

#### 2.2.3. Liposome-Mediated Plant Genetic Transformation

Liposomes are spherical vesicles composed of one or more phospholipid bilayer membranes, ranging in size from 30 nm to several μm, and composed of cholesterol and natural nontoxic phospholipids [54]. According to the size and number of bilayer membranes, liposomes can be divided into two types: multilamellar vesicles (MLV) and unilamellar vesicles. The latter is further classified into large unilamellar vesicles (LUV) and small unilamellar vesicles (SUV) [55]. Liposome-mediated transformation can introduce exogenous DNA into protoplasts through plasma membrane fusion or protoplast endocytosis (Figure 2c). Liposomes and DNA are mixed and incubated to form a DNA–lipid complex, which is subsequently mixed with protoplast suspension (supplemented with PEG), and the desired DNA is introduced into the target protoplast through liposome-protoplast fusion or endocytosis [56]. The positively charged liposome is attracted to the negatively charged DNA and the cell membrane, enabling adhesion of the liposome to the protoplast surface, followed by the incorporation of the liposome and protoplast at their binding sites, and finally releasing the plasmid into the target cells [57] (Figure 2c). To date, there are no examples of liposome-mediated genetic transformation across intact parietal cells, although liposome-mediated exogenous DNA enters protoplasts or other recipient cells [1].

#### 2.2.4. Silicon-Carbide-Whisker-Mediated Transformation

Silicon carbide whiskers (SCWs) consist of needle-like microwhiskers with a diameter of about 0.5 μm and a length of about 10–80 μm. The whiskers are tough and easily cleaved, resulting in sharp cutting edges that pierce the cell wall and eventually the cell nucleus [58]. SCW-mediated plant genetic transformation is achieved by placing suspended cells or embryogenic calli and DNA in a centrifuge tube containing SCW, which cannot bind to DNA due to its negatively charged surface [59]. Through vortexing, SCWs can create needle-like pores on the cell membrane through which exogenous DNA can enter the target cells [60,61,62] (Figure 2d). Silicon-carbide-whisker-mediated transformation is simple, fast and does not require any special instruments or equipment. However, the damage to cells during operation reduces their regeneration capacity, resulting in a relatively low conversion efficiency, and the operation process must be carried out with extreme caution due to the carcinogenic risk of silicon carbide fibers [49].

#### 2.2.5. Microinjection-Mediated Plant Genetic Transformation

Microinjection-mediated plant genetic transformation is a technique that involves injecting DNA into a single plant nucleus or cytoplasm using a glass microcapillary injection pipette [5,63]. In this technique, the target cell is fixed under a microscope; there are two micromanipulators, one of which is the holding pipette that fixes the cell and the other is a microcapillary tube containing a small amount of DNA solution to penetrate the cell membrane or nuclear membrane. Through injection, the DNA is transferred into the cytoplasm/nucleus of plant cells or protoplasts using the microcapillary pipette (0.5–10 μm at the tip), and the transformed cells are cultured and grown into transgenic plants after gene transfer is completed [64] (Figure 2e). This approach has been widely used in animal cells, but due to the thick cell wall of plants, the syringe has a difficult time effectively penetrating the cell wall and injecting exogenous DNA into the cell. However, hydrolysis of plant cell walls by hydrolase results in protoplasts’ death, which is the main obstacle in plant genetic transformation [65].

#### 2.2.6. Pollen-Tube-Pathway-Mediated Transformation

In the pollination process of higher plants, pollen forms the pollen tube after germination on the stigma surface and extends to the ovule along the style, and the pollen nucleus passes through the pollen tube to fertilize the ovule [66]. Pollen-tube-mediated plant genetic transformation entails removing the stigma from the recipient plant immediately after pollination and adding exogenous DNA solution dropwise to the recipient plant’s severed style [67]. The exogenous DNA is transported to the recipient plant’s ovary by pollen tube growth, where it is integrated with the undivided but fertilized recipient egg, resulting in the exogenous DNA being integrated into the recipient’s genome at the embryogenic stage and being present in the transformed seed [68] (Figure 2f). Pollen-tube-pathway-mediated plant transformation, unlike particle bombardment-, electroporation-, and *Agrobacterium*-mediated plant transformation, does not involve protoplast manipulation, cell culture, or plant regeneration processes, and this method-mediating DNA transfer is relatively simple, avoiding cell culture and plant regeneration processes inherent in other genetic transformation systems [69]. In addition, this method frequently avoids the drawbacks of poor regenerative ability, genotype limitation, and genetic variation such as mutation and methylation [70]. However, due to natural flowering period limits, foreign gene transformation using this approach has only been successful in a few monocot or dicot plants; therefore this method has not been widely used [23].

### 2.3. Key Factors Affecting Plant Genetic Transformation

Plant genetic transformation is a complex process that involves the transfer of target genes into plants through physical, chemical, or biological methods, followed by screening and regeneration of transgenic plants [93]. Nowadays, genetic transformation has become a common method for improving crop yield and plant traits. However, when using these methods, researchers frequently encounter issues such as gene transfer method limitation, explant/species genotype limitation, exogenous gene random integration, regeneration difficulty, and low transformation efficiency [7]. For example, *Agrobacterium*-mediated transformation is limited by the explant/species genotype [94]; the particle bombardment method can cause damage to the cell, resulting in high copy number and large-scale rearrangement of foreign DNA [23]; microinjection-mediated protoplast manipulation and culture are difficult [95]; and electroporation can destroy DNA or cause it to lose its integrity [96] (Table 3). As a result, researchers have developed a variety of strategies to optimize and improve the constraints imposed by traditional transformation approaches.

## 3. Strategies to Overcome the Limitations of Traditional Gene Transfer Methods

Traditional gene delivery methods involve inserting genes into plant cells through *Agrobacteria* or external forces (such as gene gun or electroporation). However, these methods frequently result in limitations such as plant cell damage, low transformation efficiency, and DNA integration at random sites in the genome [46]. To solve these problems, researchers have developed new strategies to transfer exogenous genes, such as the nanoparticle-mediated gene delivery method, which can deliver biomolecules to intact plant cells without the need for external forces [1,9] and co-deliver the CRISPR/Cas or ZFNs machinery into plant tissues, which mediates targeted DNA insertion at suitable genomic sites in plants [47,48].

### 3.1. Nanoparticle-Mediated Gene Delivery

Gene transfer mediated by nanoparticles can deliver biomolecules into intact plant cells without the use of external force [9], including magnetic nanoparticles, peptide nanoparticles, layered double hydroxide nanosheets (LDH), DNA nanostructures, and carbon nanotubes (Table 4).

#### 3.1.1. Magnetic-Nanoparticle-Mediated Gene Delivery

Magnetic nanoparticle-mediated transformation involves wrapping the magnetic nanoparticle (MNP) with plasmid DNA to form an MNP–DNA complex and then introducing it into pollen under the action of a magnetic field (magnetofection); the MNP–DNA–pollen complex then enters the plant through pollination and integrates into the offspring’s genome of next generations, resulting in transgenic seeds that regenerate into transgenic plants [97] (Figure 3a). Pollen magnetization has the potential to improve genetic transformation efficiency, eliminate species dependence, eliminate the regeneration process, shorten breeding times, and achieve high-throughput screening and multi-gene co-transformation, all of which are of great importance to speed up the breeding of new transgenic plant varieties [1].

#### 3.1.2. Peptide-Mediated Gene Delivery

Peptides have a low molecular weight and degradable amino acid repeats, such as cell-penetrating peptides (CPPs) and protein transduction domains (PTDs), which are both synthetic or naturally derived low molecular weight cationic and/or amphiphilic peptides [98]. CPPs are short peptides that help cells absorb small compounds, large DNA fragments, or nanoparticles [1]. PTDs are small peptides with a high basic amino acid content [99]. Initially, Rosenbluh et al. showed that fluorescently labeled histones could penetrate the plasma membrane when incubated with petunia protoplasts [100]. In peptide-mediated gene delivery, negatively charged DNA binds to CPPs (with polycations at the N-terminal), and the peptide–DNA complex enters plant cells through vacuum or compression (Figure 3b). Chang et al. demonstrated that CPPs could transmit protein to different tissues of tomato (dicots) and onion (monocots), implying that CPPs could transmit exogenous biomolecules to complete plant cells through the cell wall and lipid bilayer [101]. Subsequent studies have shown that these peptides can also deliver DNA to corn/onion root tip cells [101,102] or tomato root cells [99]. Recent studies have revealed that organelle-targeting peptides transport DNA to specific organelles in intact plants, such as the nucleus [103], mitochondria [104], and chloroplasts [105].

#### 3.1.3. Layered-Double-Hydroxide-Mediated Gene Delivery

Layered double hydroxides (LDHs) are a class of ionic layered compounds with positively charged sublayers with charge-compensating anions and solvates and an interlayer filled with charge-balancing anions and co-embedded water [106,107]. LDHs’ cationic nature allows them to bind strongly to negatively charged DNA. Bao et al. found that LDH–lactate–NS could successfully shuttle the negatively charged fluorescent dye FITC–DNA into the entire plant cytoplasm of *Arabidopsis thaliana* and tobacco (BY2) [108]. LDHs were able to deliver DNA to plant cells through three pathways: the first was through plant cell walls; however, it prevented the DNA/RNA–LDH complex from reaching the plasma membrane and cytoplasm, whereas DNA/RNA could; the second is that the DNA/RNA–LDH complex passes through the plasma membrane through a non-intracellular pathway; and the third is by the internalization of the DNA/RNA–LDH complex into plant cells via the endocytosis pathway [1] (Figure 3c).

#### 3.1.4. DNA-Nanostructure-Mediated Gene Delivery

DNA nanotechnology utilizes the base-pairing precision in DNA to assemble artificial ssDNA (single-stranded DNA) sequences into nanostructures and supramolecular structures with well-defined sizes, shapes, and geometries (including tetrahedron) by attaching different biomolecules to the cargo attachment site, such as DNA, siRNA, or protein [109]. DNA nanostructures of different sizes and shapes have now been synthesized, and they are critical for DNA, RNA, and protein drug delivery in animal systems [110] (Figure 3d). In plants, a GFP gene (which expresses constitutively in the nuclear genome) was silenced in transgenic mGFP5 tobacco (Nb) by designing a 21 bp siRNA sequence that inhibits GFP expression in a variety of monocots and dicots plants. The results showed that compared to free siRNA, loading on DNA nanostructures can effectively protect siRNA from degradation in cells, and the GFP fluorescence of all leaves soaked with siRNA loaded on DNA nanostructures is significantly reduced, indicating that DNA nanostructures can be used as an effective tool for nucleotide delivery in plant systems [1].

#### 3.1.5. Carbon-Nanotube-Mediated Gene Delivery

Carbon nanotubes (CNTs) can be divided into single-wall carbon nanotubes (SWCNTs) and multi-wall carbon nanotubes (MWCNTs). SWCNTs are made of graphene layers with cylindrical nanostructures of 0.7–3.0 nm in diameter, while MWCNTs are made of multiple SWCNTs with a diameter of 220 nm [9,111]. The DNA–CNTs complex formed by DNA and carbon nanotubes can enter the plant nucleus through the plant cell wall (Figure 3e). Liu et al. were the first to discover that SWNTs could penetrate and be internalized by the cell walls and cell membranes of intact tobacco cells [112]. Demirer et al. achieved efficient DNA transfer and high levels of protein expression in protoplasts of *Nicotiana benthamiana*, arugula, *Triticum aestivum,* and *Gossypium hirsutum*, showing that polyethyleneimine (PEI)-modified single-walled carbon nanotubes (PEI-SWCNTs) could adsorb nucleic acids through electrostatic attraction and protect them from nuclease degradation [7].

### 3.2. CRISPR/Cas/ZFN-Mediated Targeted DNA Insertion

Conventional gene transfer methods often integrate DNA at random sites in the genome, resulting in the destruction or silencing of some key functional genes, thus changing plant agronomic traits. It is an excellent option for inserting DNA into suitable genomic sites in plants [46]. Researchers are investigating a variety of methods of targeted DNA insertion in plants to obtain high efficiency and a wide range of targeted genomic sites, including CRISPR/Cas and ZFNs.

#### 3.2.1. CRISPR/Cas-Mediated Targeted DNA Insertion

The CRISPR/Cas system is composed of Cas nuclease and guiding RNA molecules that guide Cas to produce DSB (DNA double-stranded breaks) with a definite nucleotide sequence on the genome target [113]. Genome modification can be obtained by end joining or homologous recombination (gene knock-in) of exogenous donor DNA, resulting in allele replacement or targeted transgene insertion [114]. The recognition specificity can be easily changed by modifying the variable region of the guide RNA, making CRISPR/Cas a highly programmable tool [46]. This technique can be used for a variety of purposes, including targeted DNA insertion. CRISPR-induced gene knock-in of donor DNA by homology-driven repair (HDR) has been applied in maize [115], wheat [116], rice [117], *Arabidopsis* [118], and tomato [119]. The CRISPR/Cas gene and the donor DNA were introduced to plants by *Agrobacterium* or bombardment as a transgenic T-DNA locus to initiate gene targeting in plants [116,118]. After identifying the plants carrying the expected directional insertion, the original T-DNA was removed from the genome by genetic separation [46,118]. Lu et al. demonstrated that the efficiency of the chemically modified donor DNA (including phosphorylation and phosphorothioate linkages) and CRISPR/Cas9 to insert sequences into the rice genome can be improved by an order of magnitude [117].

#### 3.2.2. ZFN-Mediated Targeted DNA Insertion

Zinc finger nuclease (ZFN) is a chimeric nuclease that has the zinc finger protein DNA binding domain as well as a non-specific DNA cleavage domain [120]. Wright et al. first proved the use of ZFNs in targeted DNA insertion in plants, confirming the hypothesis that ZFNs can be used to induce homologous recombination and target DNA insertion in plants [121]. Shukla et al. used ZFNs to insert a herbicide tolerance gene into inositol-1,3,4,5,6-pentaphosphate 2- kinase (IPK1), rendering it inactive [122]. Kumar et al. developed a system in maize that simultaneously exchanges selection markers while also integrating new trait genes, allowing unlabeled trait genes to be stacked [123]. Bonawitz et al. demonstrated that a 16.2 kb DNA fragment carrying four transgenes was targeted into the soybean genome using ZFNs [48]. The successful application of ZFNs promotes the development of genetic transformation technology.

## 4. Ways to Overcome the Difficulty of Transformed Plant Regeneration

Aside from the fact that gene delivery efficiency influences transformation efficiency, the limitation of genotype dependence and low transformation efficiency caused by the limited regeneration ability during tissue culture is an urgent problem that must be solved in order to improve the efficiency of genetic transformation [10]. Plant cells are totipotent, meaning that they can form complete plants through the somatic embryogenesis pathway [133]. Since the genes involved in embryogenesis or meristem maintenance can promote somatic embryo production and bud regeneration [94], regulating and controlling the ectopic expression of plant growth- and development-related genes, including *Baby boom* (*Bbm*), *Wuschel2* (*Wus2*), and *Growth*-*regulating factor* (*GRF*), is an effective way to solve the issue with low regeneration ability after plant transformation), is important [11].

### 4.1. Baby Boom and Wuschel2

*Baby boom* (*Bbm*) and *Wuschel2* (*Wus2*) are key regulatory factors in the development of plant stem cells [134]. *Bbm* encodes an AP2/ERF transcription factor, which plays an important role in maintaining stem cells in an undifferentiated state. *Wus2* encodes homeodomain proteins, which can give the surrounding cells stem cell properties. The use of the key plant stem cell genes *Bbm*/*Wus2* in improving plant transformation efficiency has recently been reported. By manipulating the ectopic expression of the maize transcription factors *Bbm* and *Wus2*, Lowe et al. increased the efficiency of the *Agrobacterium*-mediated transformation of four monocot plants (*Zea mays* L., *Sorghum bicolor*, *Oryza sativa*, and *Saccharum officinarum*), thereby promoting direct somatic embryogenesis [11]. In this study, the *Bbm*/*Wus2* gene expression cluster of maize was used to construct the vector, and young leaves were used as explants for transformation with a transformation efficiency of 45% on average. Constitutive expression of the *Zea mays* L. *Bbm* and *Wus2* genes in maize improves transformation efficiency but results in the plants’ phenotypic and developmental alterations. A vector containing *ZmBbm*, *ZmWus*2 and green fluorescent protein (GFP) was introduced into two *Agrobacterium* strains (LBA4404 and EHA105) to infect the immature leaf segments of two *Panicum virgatum* genotypes (Summer and Blackwell), successfully producing embryogenic callus and regenerating transgenic plants through ectopic expression of *Bbm* and *Wus2* [3]. In addition, they also successfully implemented the Cre-Lox recombination system by removing the morphogenetic gene from the transgenic plants upon heat treatment of the GFP-expressing embryogenic calli, indicating that the strategy of adding and removing the morphogenetic genes allows them to transform the recalcitrant upland switchgrass [3]. Recently, Wang et al. significantly improved the efficiency of leaf-based transformation in maize and sorghum by testing different promoters that control *Wus2/Bbm* expression. Moreover, using a maize-optimized *Wus2*/*Bbm* construct, embryogenic callus and regenerated plantlets were successfully produced in eight species spanning four grass subfamilies (barley, foxtail millet, maize, pearl millet, rice, rye, sorghum, switchgrass, and teff), confirming the role of *Bbm* and *Wus2* in promoting the direct leaf-based transformation of grass species [12].

### 4.2. GRF

In addition to the widely used *Bbm*/*Wus2*, GRF is a plant-specific transcription factor that plays an important role in the development of plant leaves, stems, flowers, seeds and roots [135]. In angiosperms, gymnosperms, and mosses, the GRF transcription factor gene is highly conserved and functions in a complex with the transcription co-activator (GRF-interacting factors, GIFs) [135]. GRF is regulated by microRNA (miRNA396) at the transcriptional level, which is associated with the GRF-mediated regulation of plant growth and development [10]. In *Arabidopsis*, co-expression of *AtGRF3* and *AtGIF1* promotes leaf size development more than increased expression of these genes alone [13]. Recently, the *GRF4*-*GIF1* chimeric construct was used to produce transgenic plants with an average conversion efficiency of 65% (range in 27–96%) in two tetraploid wheat varieties (Desert King and Kronos) and 9–19% in two previously non-transformable common wheat varieties (Hahn and Cadenza). Furthermore, unlike *Bbm*/*Wus2*, overexpression of the *GRF4*-*GIF1* chimera had no negative effects on plants, and the transgenic wheat was normally fertile. Furthermore, the *GRF4*-*GIF1* chimera from citrus and grape enhanced citrus plant regeneration, indicating that *GRF4*-*GIF1* chimera overexpression broadened the range of transformable genotypes [14]. However, the limitation of this approach is that *GRF4* is negatively regulated by miR396, and destroying miR396 target sites on *GRFs* increases the *GRF* transcript, hence raising the level and activity of the GRF4-GIF1 complex [15,136]. Qiu et al. improved the efficiency of wheat regeneration and gene editing in 11 excellent wheat varieties by introducing a point mutation into the target site of miR396, which extended the genotype range that can be used for wheat transformation [10,16].

## 5. Perspectives

Plant genetic transformation can enhance crop yields and biotic and abiotic stress tolerance by imparting desirable genetic characteristics to crops [8]. By inserting specific functional genes into plants, crop traits can be significantly enhanced and plants’ ability to cope with biotic and abiotic stresses can be improved (Table 2). Transgenic sugarcane lines with a medium copy number of the *cry1Ac* gene may exhibit clear resistance to sugarcane borer, and their yield is similar or even better than non-transgenic control lines [74]. Overexpression of *Pinellia ternata* agglutinin (*ppa*) in wheat significantly improves aphid resistance [76], and overexpression of *Vitreoscilla hemoglobin* (*VHb*) in maize results in waterlogging tolerance in transgenic maize lines [72]. However, efficient genetic transformation remains a challenge for many crops [1].

*Agrobacterium*-mediated gene transfer has been successfully used in many dicots, and this method has been widely used in a large number of genotypes of rice and corn, but progress in other gramineous crops is relatively slow. Optimization of the vector [24], application of the hypervirulent *Agrobacterium* strain (AGL1), and an appropriate selection marker (*bar* gene) improved the efficiency of *Agrobacterium*-mediated transformation efficiency in monocot plants (Table 1). By selecting suitable explants and strains, optimizing the co-culture system, and screening transformants, *Agrobacterium*-mediated transformation can be applied to more crops. In addition, *Agrobacterium*-mediated transformation has been employed to produce transgenic crops without selective markers [28,137,138]. Because no commercial transgenic crop varieties have been developed, its transformation efficiency is substantially lower than that of other species and has a negative attitude towards transgenic plants. However, it is considered important to produce marker-free cultivars if the marker genes used to produce positive transgenic plants are eliminated [28]. Marker-free transgenic plants have been generated with the use of an *Agrobacterium*-mediated co-transformation system using a plasmid containing two independent T-DNA regions for many species including wheat [28], rice [138], sorghum [139], and soybean [137]. Transgenic crops produced by *Agrobacterium*-mediated transformation of marker-free genes may eliminate potential environmental and biosafety issues. To improve existing cultivars and generate new excellent cultivars, it is desirable to improve existing strategies and develop new methods of plant genome manipulation [140].

Unlike *Agrobacterium*-mediated transformation, particle-bombardment-mediated genetic transformation is not limited to receptor materials. While enabling early monocot transformation, it will also result in high copy number and large-scale rearrangement of foreign DNA, as well as multicopy inserts that will lead to transgene silencing [23]. With the in-depth understanding of DNA repair mechanisms (including HR and NHEJ), targeted DNA insertion at suitable genomic sites in plants is an ideal alternative method [46]. Particle bombardment has been used to deliver the CRISPR/Cas or ZFNs system and the repair template to plant tissues, as well as to achieve targeted insertion of selective marker genes in rice [47] and soybean [48].

Traditional methods of gene introduction are used to introduce genes into plant cells by *Agrobacterium* or external force (such as gene gun or electroporation). However, these methods have some limitations, such as plant cell damage, multiple copies, and DNA integration at random sites in the genome [46]. Nanoparticles are substances with highly adjustable physicochemical properties that can pass through the plants’ cell wall without the use of any external force [8]. In contrast to conventional *Agrobacterium-* and particle-bombardment-mediated gene delivery, nanoparticles have the advantages of low cytotoxicity, ease of handling (for example, no need for cell wall removal or expensive equipment), wide host range suitability, and the ability to deliver a variety of biomolecules (nucleic acids, protein, and regulatory active molecules) [1,8], making them useful as effective tools for biomolecules’ delivery into plants. Nanoparticle-mediated genetic transformation has been successfully applied to various species (Table 4).

Although some progress has been made in nanoparticle-mediated gene delivery, as an emerging biotechnology field, its development time is limited, and many issues remain. First, due to the cell wall, the delivery of nano-carriers into plants has not been thoroughly investigated. Without external assistance, it is necessary to understand how nanomaterials are internalized into plant cells in order to logically design them for future applications in plant biotechnology [129,141]; second, nanoparticle carriers’ design is a complex multivariable optimization process, and successful nanocarriers may need to be tailored to different plant systems until a complete nanoparticle delivery system for the plant system is established [8]; and finally, some nanoparticle-mediated transformation does not result in transgenic plants, and plant regeneration is the greatest challenge with nanoparticles for genetic transformation [9]. Some studies have shown that ectopic expression of some MTFs can regulate the regeneration of newly transformed tissues [11,12,16]. The combination of nanoparticle-mediated genetic transformation and overexpression of MTFs has the potential to make significant advances in the field of plant genetic engineering.

Ectopic expression of MTFs promotes organogenesis or direct somatic embryogenesis in a variety of monocotyledons and dicotyledons, significantly improving regeneration/transformation efficiency, expanding the range of substitute explants for transformation, and accelerating the transformation process. However, it is not possible to use a single MTF or a combination of transcription factors to universally transform all recalcitrant species or genotypes. There are still genotypic-dependent differences in response to different MTFs and further modifications are required to increase the frequency of transformation [10]. In addition to the genes mentioned above, many other upstream and downstream interaction factors that promote meristem formation, bud regeneration, or somatic embryogenesis for plant transformation have yet to be discovered [142]. Future research could focus on determining the synergistic and additive effects of various combinations of different growth and development regulatory genes on plant transformation; also, fine-tuning the expression of these genes is critical for the regeneration of normal fertile plants from different plant species.

## Figures and Tables

**Figure 1 ijms-24-10646-f001:**
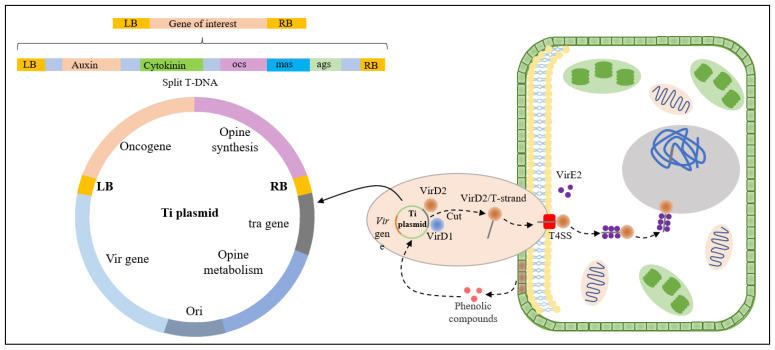
Schematic diagram of T-DNA transfer and integration into the plant genome. VirA, VirG: perception of phenolic compounds from plant wounds/induction of *virulence* (*Vir*) gene expression. VirD1: DNA topoisomerase processing T-DNA. VirD2: Endonuclease cutting the T-DNA border to initiate T-strand synthesis and attached to 5′ of T-strand/formation of T-DNA complex/transport of the T-DNA complex through nuclear pores. VirE1: Plays the role of a chaperone to stabilize VirE2 in *Agrobacterium*. VirE2: Single-strand DNA binding protein protecting the T-strand from nuclease. T4SS: Type IV secretion system.

**Figure 2 ijms-24-10646-f002:**
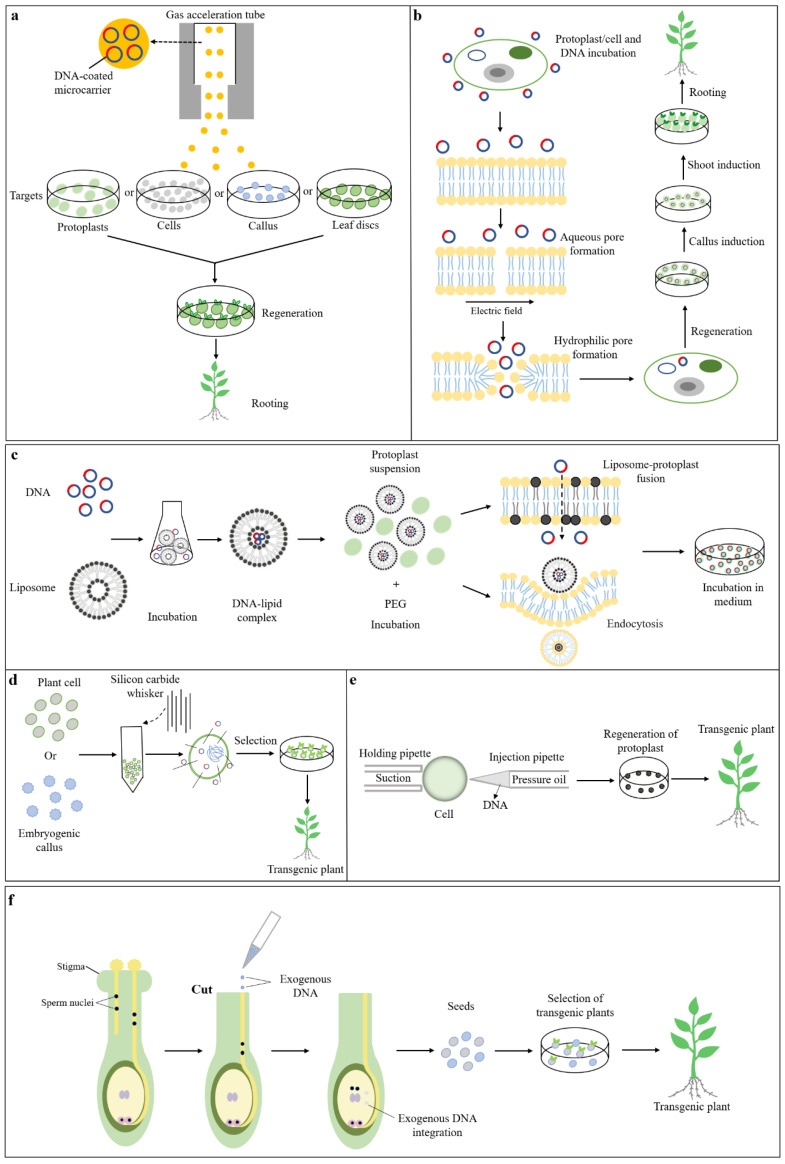
Schematic diagram of different genetic transformation mediated by the direct method. (**a**) Particle-bombardment-mediated plant transformation. (**b**) Electroporation-mediated transformation. (**c**) Liposome-mediated transformation. (**d**) Silicon-carbide-whisker-mediated transformation. (**e**) Microinjection-mediated transformation. (**f**) Pollen-tube-pathway-mediated transformation.

**Figure 3 ijms-24-10646-f003:**
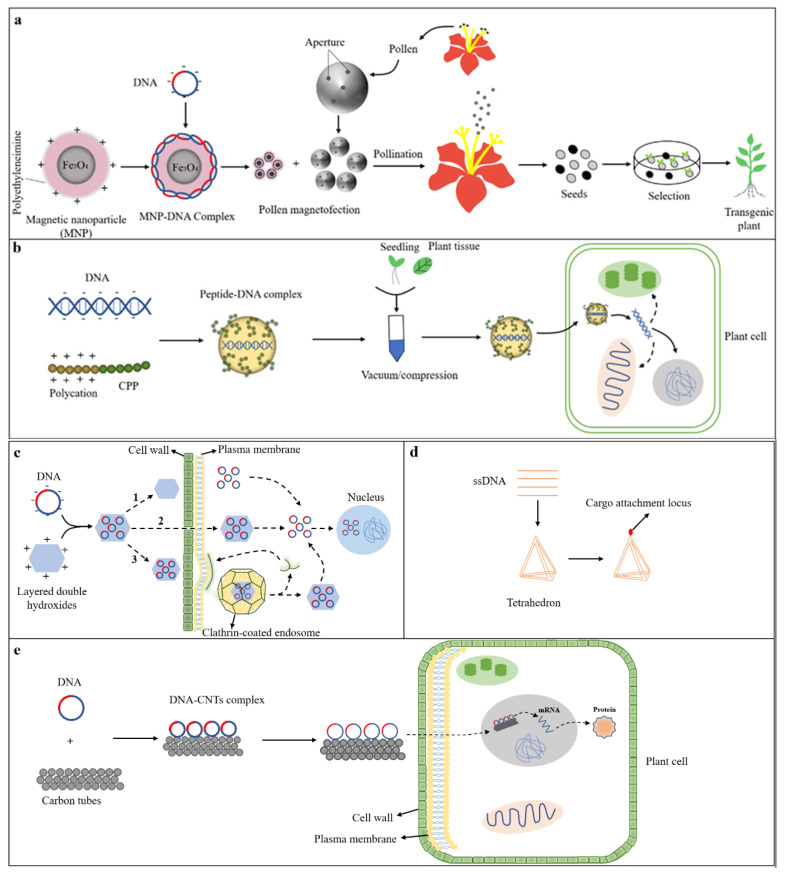
Nanoparticle-mediated gene transfer: (**a**) magnetic-nanoparticle-mediated gene transfer; (**b**) peptide-nanoparticle-mediated gene transfer; (**c**) layered-double-hydroxide-nano-transporter-mediated gene transfer; 1. plant cell walls pathway; 2. plasma membrane by a non-intracellular pathway; 3. endocytosis pathway; (**d**) DNA-nanostructure-mediated gene transfer; (**e**) carbon-nanotube-mediated gene transfer.

**Table 1 ijms-24-10646-t001:** *Agrobacterium*-mediated transformation of monocots and dicots.

Species	Explants	Genotype	Target Gene	*Agrobacterium* Strain	Vector	Selectable Marker	Efficiency (%)	Reference
Monocots								
Wheat	Immature embryos	Bobwhite SH98 26	*sGfp*	AGL1	pGH215	*Hpt*	Up to 15%	[26]
	Immature embryos	Fielder	*gus*	AGL1	pGoldenGreenGate	*Hpt*	Up to 25%	[25]
	Immature embryos	CB037, Fielder, Jimai 22 Kenong 199, Shi 4185	*gus*	C58C1	pRK2013	*Bar*	2.8–53%	[28]
	Mature and immature embryos	DBW 88, DBW 90, DBW 93, DPW 621-50, HD 3086 and WH 1105	*gus*	EHA105	pCAMBIA3301	*Bar*	9.8–14.9%	[27]
Rice	Immature embryos	a broad range of species	*gfp*	LAB4404 and EHA105	pPUG1-1	*Hpt*	-	[29]
	Embryogenic calli	Sambha mahsuri	*AmSOD1*	LAB4404 and EHA105	pSFSOD1	*Hpt*	-	[30]
Maize	Immature embryos	ND101 and ND88	*DsRed*	EHA105		*Bar*	Up to 17.6%	[31]
	Immature embryos	HC69 and PH2RT	*YFP*	LBA4404THY-	pVIR	PAT/PMI	-	[24]
Barley	Immature embryos	Scarlett and Golden Promise	*HvCKX2*	AGL1	pMCG161	*Bar*	3.47%	[32]
	Immature embryos	Golden Promise	*gus*	AGL1	pBRACT	*Hpt*	25%	[33]
Dicots								
*Eucalyptus*	Leaves	E. urophylla × E. grandis clone DH32- 29	*gus*	GV3101, LBA4404, EHA105, and AGL1	pBI121	*Npt II*	1.9%	[34]
Poplar	Leaves	*Populus Alba*×*Populusglandulosa* Uyeki	*gus*	GV3101	35S:GUS vector	-	-	[35]
	Callus	84K	*gus*	GV3101	pCAM BIA1301	Hygromycin B	greater than 50%	[36]
*Codonopsis pilosula*	Stems	(Franch.) Nannf.	*gus*	GV3101	pCAMBIA1381	*Hpt*	91.07%	[37]
Tea	Callus	*Camellia* *sinensis*	*gus*	EHA105	PS1aG-3	-	3.6%	[38]
*Liriodendron hybrid*	Callus	52053	*gus*	EHA105	pBI121	Geneticin	60.7%	[39]
Pigeon pea	Cotyledons	ICPL 85063	*gus/gfp*	LBA4404	pCAMBIA1301	*Hpt*	83%	[40]
Soybean	Cotyledonary node	Jack	*GsWRKY20*	EHA101	myc-pBA	*Bar*	-	[41]
*Ailanthus altissima* (Mill) Swingle	Shoots	-	*gfp*	K599	pCAMBIA1300	-	-	[21]
Cotton	Shoot apical meristem	*Gossypium hirsutum*, *Gossypium barbadense* and *Gossypium* *arboreum*	*GFP* and *RUBY*	-	pCAMBIA 2300	*AADA*	Up to 8.01%	[42]

**Table 2 ijms-24-10646-t002:** Direct transformation methods.

Species	Explants	Genes/Molecules	Gene Delivery System	Efficiency (%)	Reference
Barley	Seeds	*OsWRKY70, OsWRKY53,* and *gus*	Particle bombardment	-	[71]
Maize	Calli	*VHb*	Particle bombardment	-	[72]
Sorghum	Immature embryos	*NptII*	Particle bombardment	46.6%	[73]
Sugarcane	Embryonic calli	*cry1Ac* and *bar*	Particle bombardment	-	[74]
Rice	Calli	Cpf1, crRNA, and repair templates	Particle bombardment	8%	[47]
Wheat	Immature embryo	*gfp* and *bar*	Particle bombardment	-	[75]
Wheat	Callus	*Ppa*	Particle bombardment	-	[76]
Palm	Callus	*ChoA*	Particle bombardment	-	[77]
Blackgram	Embryonic axis	*ChiB*	Particle bombardment	13%	[78]
Cowpea	Embryonic axis	*Arc1*	Particle bombardment	-	[79]
Carrizo citrange	Immature epicotyl	*nptII* and *gfp*	Particle bombardment	18.4%	[80]
Soybean	Embryogenic cells	*Hpt,* ZFN expression constructs, and HDR donor	Particle bombardment	About 2.84%	[48]
Zygnematophycean algae	Cells	*GFP*	Electroporation	-	[81]
Wheat	Immature embryos	*bar* and *uidA*	Electroporation	0.4%	[82]
Tomato	Leaves	Fe and Mg	Liposomes	33%	[83]
Maize	Embryogenic callus	*gus* and *bar*	Silicon carbide whisker	-	[84]
Cotton	Embryogenic callus	*Gus* and *AVP1*	Silicon carbide whisker	Up to 94%	[85]
Cotton	Embryogenic callus	*GUS, AVP1,* and *nptII*	Silicon carbide whisker	-	[86]
Peanut	Callus	chitinase and *hygromcin*	Silicon carbide whisker	6.88%	[58]
Barley	Protoplasts	*Act1, gus,* and *nos*	Microinjection	-	[87]
Oil palm	Protoplasts	*GFP*	Microinjection	10–74.6%	[88]
Maize	Pollen	*GFP*	Pollen tube pathway	0.86%	[89]
Cotton	Pollen	*nptII*	Pollen tube pathway	-	[90]
Melon	Pollen	*Fom-2*	Pollen tube pathway	3.28% and 4.26%	[91]
Peanut	Pollen	*AhBI-1*	Pollen tube pathway	50%	[92]

**Table 3 ijms-24-10646-t003:** Comparison of different plant genetic transformation methods.

Transformation Methods	Tissue Type	Species	Delivery Type	Advantages	Disadvantages
*Agrobacterium*	Cells, tissues, and whole plants	Monocot and dicot	DNA	It has high transformation efficiency and stability	It is species and genotype restricted, and random integration may result in gene destruction
Particle bombardment	Any intact tissue or explant	Monocot and dicot	DNA, siRNA, miRNA, and RNP	It is not limited by tissues or cell types	The transferred DNA is not protected, the transformation efficiency is lower than with *Agrobacterium*-mediated transformation, and the equipment used is costly. High copy numbers and extensive rearrangements of foreign DNA, as well as the integration of multiple copies of the same gene in the genome, often lead to gene silencing
Electroporation	Leaf blade, protoplast, meristem, and pollen grain	Green algae, monocot and dicot	DNA, siRNA, miRNA, and protein	It is possible to transform whole cells and tissues. The transformation efficiency depends on the plant’s material	It requires cell wall removal and is limited to an in vitro suspension system. It will cause tissue damage without specificity, and transformed cells have a 50% chance of survival
Liposome	Protoplast, callus, and pollen	Dicot	DNA, RNA, and protein	The wrapped nucleic acid can be protected from degradation by nucleases; specific cells as well as various cell types can be targeted.	Its transformation efficiency is low
Silicon carbide whisker	Callus and mature embryos	Monocot	DNA	Its operation is simple, and its cost is low	It has a low transformation efficiency, and silicon carbide whiskers are toxic
Microinjection	Protoplasts, immature embryos, and pollen	Monocot and dicot	DNA	The method, which is technically simple, may facilitate the transfer of genes to grains that are not easily regenerated from cultured cells	Its transformation efficiency and frequency are low, it takes a long time to complete, it is costly, and it requires trained and certified workers to conduct experiments
Pollen tube	Pollen tube	Monocot and dicot	DNA	It does not involve tissue culture or in vitro regeneration	Its transformation efficiency is low, and the transfer of exogenous genes is limited by natural flowering period

**Table 4 ijms-24-10646-t004:** Nanoparticle-mediated transformation.

Species	Explants	Molecules	Nanoparticles	Efficiency (%)	Reference
Maize	Pollen	*RFP*, *GUS,* and *EGFP*	Magnetic nanoparticles	32–55% (DNA entry)	[124]
Maize	Immature embryos	Cre recombinase Protein	Magnetic nanoparticles	20% (bombarded embryos produced calli with the recombined loxP sites)	[125]
Okra	Embryo	*mgfp*	Magnetic nanoparticles	-	[126]
Cotton	Pollen	*BTΔα-CPTI,* and *GUS*	Magnetic nanoparticles	About 1%	[97]
Rice	Calli	*aadA* and *gfp*	Peptide nanoparticles	About 2.77%	[127]
Kenaf	Cotyledon or calli	*aadA* and *gfp*	Peptide nanoparticles	About 0.037%	[127]
Tomato	Pollen	dsRNA	Layered double hydroxides	-	[128]
*Nicotiana* *benthamiana*	Leaves	siRNAs	DNA nanostructures	-	[129]
Rice	Leaves and excised embryo	*GFP*, *YFP,* and *GUS*	Carbon nanotubes	-	[130]
Wheat	Leaves	*sGFP*	Carbon nanotubes	-	[7]
*Nicotiana* *benthamiana*	Leaves	*sGFP*	Carbon nanotubes	-	[7]
Cotton	Leaves	*sGFP*	Carbon nanotubes	-	[7]
*Nicotiana* *Benthamiana*	Leaves	siRNA	Carbon nanotubes	95% (gene silencing rate within 24 h)	[131]
*Arabidopsis thaliana*	Seedlings	*GFP* and *RLuc*	Carbon nanotubes	-	[132]

## Data Availability

Not applicable.

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
