# Peer review of "Technological Development and Application of Plant Genetic Transformation"

_ijms, 2023, doi:10.3390/ijms241310646_

Round 1
Reviewer 1 Report
The authors provide an overview of plant genetic transformation techniques. A number of – also recent – reviews have been published on this topic, so the novelty of the topic is limited. However, the authors included two aspects that may be interesting to the readers, namely nanoparticles-mediated gene delivery and the discussion on transcription factors (in context with the “Perspectives” section 4.) In order to add to the value of the manuscript to the readers, it will be necessary to set the focus on aspects that are not included in other reviews.
Please clarify for each table showing examples of references what the rationale of including references was, and explain the table contents in the captions. It would be beneficial for the manuscript if the focus was on the most recent publications, unless you would like to show something else. In the current tables, it seems like a random collection of papers. After having reflected on this, please critically review the references.
Please check and rephrase 2.1.; bring it in context with Figure 1. In my opinion, too many details that are moreover not explained in the manuscript are provided in Figure1, whereas the text can be improved by adding relevant details that can be mirrored in the Figure. Also, I find the caption of Figure 1 not convincing, e.g., the sentence mentioning T4SS likes context and T4SS is not explained anywhere. Please check both text and Figures for which details are essential for the reader and otherwise refer to reviews that show this transformation method in a comprehensive manner. What does Table 1 focus on? Please bring everything in context and focus on novelties (e.g. also, if applicable, the later discussed transcription factors).
Which strategies may overcome the limitations experienced with monocot transformation using Agrobacteria? What has led to the successful transformation of the monocots mentioned in Table 1? The aspect of dicots and monocots may need more attention and could be discussed.
Line 108: Which species? Bring in line with Table 1.
2.2. Please critically review whether Figure 2 supports the text and vice versa.
Figure 2: I suggest to not use the abbreviation SCW in the figure or explain it. The Figure caption should contain some explanatory text or use the text body and refer to Figure 2. Figure 2.f. may contain unnecessary information that is not relevant for explaining the technique.
2.2.1.: Lines 148-149: Why should researchers investigate further and is this reasonable? Please explain in the text why and for which purpose.
2.2.6.: It is important to also explain why this technique is not the general method of choice. Currently you basically list only advantages, and as I understood one of your major discussion points is the regenerative capacity and how it can be enhanced. This method would overcome these limitations.
3. You may think about putting 3. – Key factors – at the end of the explanation of transformation methods.
3.1. may then be a completely new paragraph that explains one solution to the challenges. Please think about bringing the challenges and how they can be solved in a systematic context to your explanations before.
Also here, please critically go through text, figure, and the table, explain and bring into context as mentioned before for the other paragraphs.
Similarly, for the plant regeneration: Please bring into a systematic context with the transformation methods described before.
Line 64: should be WUSCHEL2
Line 137: specify eluted.
Please check the sentences - some could be shortened to improve readability.
Please check for typos.
Author Response
Thank you very much for your time and efforts in handling and reviewing our manuscript. Your insightful comments and suggestions greatly helped us in improving the quality of the manuscript. We revised the manuscript according to your factful and valuable comments and suggestions, and tried to improve manuscript’s quality by providing point-by-point responses as follows;
Point 1
Please clarify for each table showing examples of references what the rationale of including references was, and explain the table contents in the captions. It would be beneficial for the manuscript if the focus was on the most recent publications, unless you would like to show something else. In the current tables, it seems like a random collection of papers. After having reflected on this, please critically review the references.
Response 1
Thank you for your valuable comments and your patience with our manuscript. The tables were revised. Table 1 is grouped according to Agrobacterium-mediated transformation in monocots and dicots. Table 2 is classified according to the transformation mediated by different direct transformation methods in monocots and dicots. The contents of the table are explained in the captions, and the references have been replaced by new publications.
Point 2
Please check and rephrase 2.1.; bring it in context with Figure 1. In my opinion, too many details that are moreover not explained in the manuscript are provided in Figure1, whereas the text can be improved by adding relevant details that can be mirrored in the Figure. Also, I find the caption of Figure 1 not convincing, e.g., the sentence mentioning T4SS likes context and T4SS is not explained anywhere. Please check both text and Figures for which details are essential for the reader and otherwise refer to reviews that show this transformation method in a comprehensive manner. What does Table 1 focus on? Please bring everything in context and focus on novelties (e.g. also, if applicable, the later discussed transcription factors).
Response 2
Thank you for pointing this out. 2.1 text has been rephrased, unnecessary figures and texts have been deleted, and the relevant details from Figure1. have been supplemented in the revised manuscript (Page 2 and 3, line 92-102).
Table 1 has been updated to compare different strains, selectable markers, vectors and Agrobacterium-mediated genetic transformation efficiency in monocots and dicots.
Point 3
Which strategies may overcome the limitations experienced with monocot transformation using Agrobacteria? What has led to the successful transformation of the monocots mentioned in Table 1? The aspect of dicots and monocots may need more attention and could be discussed.
Response 3
Thanks you very much for pointing this out. We incorporated your suggestion into the manuscript as follows:
-“The breakthrough in monocots plant transformation by Agrobacterium comes from a better understanding of the key factors or parameters required for effective grain infection and gene transfer, such as using explants with a large portion of active dividing cells, which refers to immature embryos. Also it has been found that employing highly toxic Agrobacterium strains and suitable vectors are necessary for Agrobacterium to successfully transform cereal (Page 3, line122-135)”.
-The methods used to successful transform monocots listed in Table 1 employ immature embryos with high division cells activity and selectable markers suitable for monocots such as bar. In addition, by utilizing hypervirulent strains such as AGL1, the transformation efficiency of monocots mediated by Agrobacterium can be improved.
-5. Perspectives (Page 25, line 519-526) addressed the aspect of dicots and monocots.
Point 4
Line 108: Which species? Bring in line with Table 1.
Response 4
Thank you for the nice reminder. The species were consistent with Table 1 (Reference 22).
Point 5
2.2. Please critically review whether Figure 2 supports the text and vice versa.
Response 5
Thanks for your comments. We have critically revised Figure 2 and the corresponding text, removed unnecessary content and added annotation information to Figure 2 (Page 10).
Point 6
Figure 2: I suggest to not use the abbreviation SCW in the figure or explain it. The Figure caption should contain some explanatory text or use the text body and refer to Figure 2. Figure 2.f. may contain unnecessary information that is not relevant for explaining the technique.
Response 6
Thank you for your suggestion. Accordingly, we have included the full name of SCW in Figure 2d, and we have removed unnecessary information that is not relevant to explain the technology in Figure 2.f.
Point 7
2.2.1.: Lines 148-149: Why should researchers investigate further and is this reasonable? Please explain in the text why and for which purpose.
Response 7
Thank you for your comment. There was an oversight in the previous statement, and we have revised it. The reason why researchers should investigate further has been explained in the text as follows:
“Researchers are investigating potential mechanisms for these complexities and seeking solutions, and targeted DNA insertion at suitable genomic sites in plants is a promising alternative [47]. (Page 9, line 189-194)”.
Point 8
2.2.6.: It is important to also explain why this technique is not the general method of choice. Currently you basically list only advantages, and as I understood one of your major discussion points is the regenerative capacity and how it can be enhanced. This method would overcome these limitations.
Response 8
Thank you for pointing this out.
The reasons why this technique is not the general method of choice have been supplemented in the text as follows:
“However, the natural flowering period limits foreign gene transformation, and this approach has only been successful in a few monocot or dicot plants, therefore this method has not been widely used [24]. (Page 13, line 285-287)”.
Point 9
- You may think about putting 3. – Key factors – at the end of the explanation of transformation methods.
Response 9
Thanks for your comments. Accordingly, 3. - Key factors - has been moved to 2.3 and added at the end of the explanation of transformation methods (Page 16, line 289-303).
Point 10
3.1. may then be a completely new paragraph that explains one solution to the challenges. Please think about bringing the challenges and how they can be solved in a systematic context to your explanations before.
Response 10
Thank you very much for your suggestion.
3.1. has been moved to a new paragraph (3) that explains how to overcome the challenges of traditional gene transfer methods (Page 17, line 305-317), and 3.2, which introduces the targeted gene insertion strategy mediated by CRISPR/Cas and ZFNs, has been added (Page 21, line 406-441).
Point 11
Also here, please critically go through text, figure, and the table, explain and bring into context as mentioned before for the other paragraphs.
Response 11
Thank you very much for your nice reminder. We did our best to revise the text, figure, and table, as well as explain and clarify the other paragraphs.
Point 12
Similarly, for the plant regeneration: Please bring into a systematic context with the transformation methods described before.
Response 12
Thanks for your comments. For plant regeneration, a new paragraph 4 was added to explain the ways to overcome the difficulty of plant regeneration (Page 24, line 443-453). 3.2 was updated to 4.1 to introduce Baby boom and Wushcel2 (Page 24 and 25, line 454-483), and 4.2 mainly introduced GRF (Page 25, line 484-505).
Point 13
Line 64: should be WUSCHEL2
Response 13
Thank you very much for the reminder. We have made revisions accordingly.
Point 14
Line 137: specify eluted.
Response 14
Thanks for your comments. We apologize for this oversight. We revised this sentence as follows:
“High-pressure helium pulses accelerate DNA-coated microcarriers into the gas acceleration tube using an electric discharge or a pressurized helium gas stream. These particles gain sufficient momentum to pierce recipient cells at high speed, while the target gene coated on the outside remains in the cell (Page 6, line 169-172)”.

Reviewer 2 Report
The review “Technological Development and Application of Plant Genetic Transformation” discusses the technological development and application of genetics in plants. It explores how genetic transformation can be an important strategy for improving plant performance. The text also presents various genetic transformation technologies and compares them. However, there is no specific information about environmental benefits or application in forestry. The microinjection method is mentioned as a simple technique, but with low transformation efficiency and frequency, as well as being time-consuming and expensive.
However, some additional points that could be addressed regarding the subject include:
- The ethical and social implications of genetic transformation in plants, such as food safety and intellectual property of genetically modified seeds.
- The use of emerging techniques such as CRISPR/Cas9 gene editing to improve the precision and efficiency of genetic transformation in plants.
- The application of genetic transformation in plants for specific purposes such as biofuel or medicine production.
- The challenges faced in the regulation and approval of genetically modified crops in different countries and regions around the world.
The authors should critically discuss the existing literature, point out the knowledge gaps, and suggest further research. The manuscript slightly lacks coherence in storyline, and English language also needs careful editing for better readability.
The review, however, a little must be improved in terms of writing since some grammar errors are present in the manuscript. The manuscript needs restructuring and substantially revised.
Author Response
Thank you very much for your time and efforts in handling and reviewing our manuscript. Your insightful comments and suggestions greatly helped us in improving the quality of the manuscript. We revised the manuscript according to your factful and valuable comments and suggestions, and tried to improve manuscript’s quality by providing point-by-point responses as follows;
Point 1
However, some additional points that could be addressed regarding the subject include:
- The ethical and social implications of genetic transformation in plants, such as food safety and intellectual property of genetically modified seeds.
Response 1
Thank you for this comment and your patience with our manuscript. The social implications of genetic transformation in plants have been discussed in 5. Perspectives as follows:
“Because no commercial transgenic crop varieties have been developed, its transformation efficiency is substantially lower than that of other species, and the public has a negative attitude towards transgenic plants. However, it is considered important to produce marker-free cultivars if the marker genes used to produce positive transgenic plants are eliminated [29]. (Page 25, line 527-530)”.
Point 2
- The use of emerging techniques such as CRISPR/Cas9 gene editing to improve the precision and efficiency of genetic transformation in plants.
Response 2
Thank you very much for your comment. The use of emerging techniques such as CRISPR/Cas9 gene editing to improve the precision and efficiency of genetic transformation in plants has been supplemented in the revised manuscript (Page 21, line 405-428).
Point 3
- The application of genetic transformation in plants for specific purposes such as biofuel or medicine production.
Response 3
Thank you very much for your remark. The application of genetic transformation in plants for specific purposes has supplemented in 5. Perspectives (Page 25, line 508-515).
Point 4
- The challenges faced in the regulation and approval of genetically modified crops in different countries and regions around the world.
Response 4
Thank you for your suggestion. As this manuscript mainly focuses on existing plant genetic transformation technology and its limitations, it does not address the challenges faced by the regulation and approval of genetically modified crops in different countries and regions around the world. However, at 5. Perspectives (Page 25 and 26, line 525-537), we included the technology of transgenic crops with markers-free mediated by Agrobacterium which may make the challenge of supervision and approval of genetically modified crops in different countries less arduous.
Point 5
The authors should critically discuss the existing literature, point out the knowledge gaps, and suggest further research. The manuscript slightly lacks coherence in storyline, and English language also needs careful editing for better readability.
Response 5
Thank you very much for your remark. We have critically discussed the existing literature, identified knowledge gaps, and suggested further research in 5. Perspectives section. We also went through the entire manuscript to eliminate grammatical mistakes.

Reviewer 3 Report
The topic of the review is of great interest to the researcher in plant genetic transformation nevertheless some minor issues should be revised before publication. The authors explain the different methods for plant genetic transformation without citing the innovative approaches used in each of them. The innovative approach is the use of nanoparticles that needs, as the same authors reported, to be more investigated. Regarding the Gen Gun method, the authors sustain that is independent of the genotype (page 6 lane 134, and Table 3), but this is not completely true because the “damage to the cell” cited on page 11 lane 246 largely depends on the genotype and specie used. This is confirmed by the low number of species transformed by this method and reported in Table 2. In my opinion, some references cited by the authors do not completely fit with the text and in some cases are too old to be cited in a review.
Page 2 lane 82: I suggest using the term “vector” instead of “medium”
Page 6 lane 141: “Plant genetic transformation mediated by this method…”, is not clear
Page 8 lane 179: “In this technique…”, the sentence is not clear, maybe is too long
Page 13 lane 301: “The result show that, when comparing….”, other than the use of the verb in the third person singular (shows), the sentence is too long and difficult to understand clearly.
The English grammar is fine, but some sentences need to be clarified.
Author Response
Thank you very much for your time and efforts in handling and reviewing our manuscript. Your insightful comments and suggestions greatly helped us in improving the quality of the manuscript. We revised the manuscript according to your factful and valuable comments and suggestions, and tried to improve manuscript’s quality by providing point-by-point responses as follows;
Point 1
The topic of the review is of great interest to the researcher in plant genetic transformation nevertheless some minor issues should be revised before publication. The authors explain the different methods for plant genetic transformation without citing the innovative approaches used in each of them. The innovative approach is the use of nanoparticles that needs, as the same authors reported, to be more investigated. Regarding the Gen Gun method, the authors sustain that is independent of the genotype (page 6 lane 134, and Table 3), but this is not completely true because the “damage to the cell” cited on page 11 lane 246 largely depends on the genotype and specie used. This is confirmed by the low number of species transformed by this method and reported in Table 2. In my opinion, some references cited by the authors do not completely fit with the text and in some cases are too old to be cited in a review.
Response 1
We regret for this oversight and thank you very much for your remarks as well as your patience with our manuscript.
-The sentence “Compared to Agrobacterium-mediated plant genetic transformation…” have been corrected as follows:
“Particle bombardment-mediated plant transformation is not limited to the source of receptor materials; cells, calli, immature embryos and organs can all be used as targets for transformation (Table 2). (Page 8, line 162-164)”.
-References were updated and new references were added accordingly.
Point 2
Page 2 lane 82: I suggest using the term “vector” instead of “medium”
Response 2
Thank you very much for your suggestion. The term “medium” has been replaced with the term “vector”.
Point 3
Page 6 lane 141: “Plant genetic transformation mediated by this method…”, is not clear
Response 3
Thank you for your comment. We have revised the sentence as follows:
“Plant genetic transformation mediated by particle bombardment is distinguished by the diversity of target materials and ease of operation (Page 8, line 176-177)”.
Point 4
Page 8 lane 179: “In this technique…”, the sentence is not clear, maybe is too long
Response 4
Thank you for pointing this out. The sentence has been revised as follows:
“Liposome-mediated transformation can introduce exogenous DNA into protoplasts through plasma membrane fusion or protoplast endocytosis. Liposomes and DNA are mixed and incubated to form DNA-lipid complex, which is subsequently mixed with protoplast suspension (supplemented with PEG), and the desired DNA is introduced into the target protoplast through liposome-protoplast fusion or endocytosis [57]. (Page 12, line 226-231)”.
Point 5
Page 13 lane 301: “The result show that, when comparing….”, other than the use of the verb in the third person singular (shows), the sentence is too long and difficult to understand clearly.
Response 5
Thank you very much for your suggestion. Accordingly, the sentence was revised as follows:
“The results showed that compared to free siRNA, loading on DNA nanostructures can effectively protect siRNA from degradation in cells, and the GFP fluorescence of all leaves soaked with siRNA loaded on DNA nanostructures is significantly reduced, indicating that DNA nanostructures can be used as an effective tool for nucleotide delivery in plant systems [1]. (Page 18, line 375-380)”.
